# Effect of *Pleurotus eryngii* on the Characteristics of Pork Patties during Freezing and Thawing Cycles

**DOI:** 10.3390/foods13030501

**Published:** 2024-02-05

**Authors:** Miaojing Zhang, Yangyang Chai, Fangfei Li, Yihong Bao

**Affiliations:** 1College of Life Sciences, Northeast Forestry University, Harbin 150040, China; zmiaojing24@163.com (M.Z.); chaiyang824@163.com (Y.C.); 2Key Laboratory of Forest Food Resource Utilization in Heilongjiang Province, Northeast Forestry University, Harbin 150040, China

**Keywords:** pork patties, freezing-thawing cycles, characteristics, *Pleurotus eryngii*

## Abstract

Temperature fluctuations severely damage the quality, oxidation stability, and structure of pork patties. This study investigated the potential reasons for *Pleurotus eryngii* (Pe) to protect frozen pork patties from quality degradation caused by temperature fluctuations and promoted the application of a natural ingredient. In this experiment, the pH, the water holding capacity (WHC), the properties of color and texture, the appearance, the degree of protein and lipid oxidation, and the microstructure of patties with different additions of Pe (0%, 0.25%, 0.50%, 1.00%, and 2.00%) were intensified during freezing and thawing (F–T) cycles. The results showed that patties with 0.50% Pe exhibited a distinguishable improvement in the changes of pH, WHC, color, and texture during F–T cycles (*p* < 0.05). With the times of F–T cycles increasing, 0.50% Pe was able to inhibit lipid oxidation of patties by decreasing the peroxide value (POV) and the thiobarbituric acid reactive substances (TBARS) value to 0.87 and 0.66-fold, respectively, compared to those in the control group. It was also able to suppress the protein oxidation of the patties with a protein sulfhydryl content increasing to 1.13-fold and a carbonyl content decreasing to 0.49-fold compared to the patties in the control group (*p* < 0.05) after 5 F–T cycles. In addition, the figures of appearance and microstructure of samples indicated that 0.50% Pe effectively restrained the deterioration of structure features from patties after 5 F–T cycles. Thus, the addition of Pe effectively maintained the characteristics of pork patties under F–T cycles.

## 1. Introduction

The pork patty is an important material in fast food production, like the hamburger. In order to preserve the essential nutrients and prolong its shelf life, frozen storage plays a critical role in fighting against lipid and protein oxidation of patties. However, the degradation degree of meat quality will increase because of oxidation and microbial growth during the frozen storage and processing of patties. During the frozen storage process, it is common for producers, sellers, and consumers to repetitively thaw and freeze pork patties [1]. The incomplete cold chain transportation systems or changes in transportation environments exacerbate the quality deterioration of pork patties, resulting in things like excessive water loss, darkening of meat color, deterioration of flavor, and decrease in tenderness [2]. There are two main reasons why pork patties tend to be spoiled. Firstly, the formation and recrystallization of ice crystals causes destruction of the cell tissue, water mobility, and redistribution, which decreases the quality of the patty [3]. Secondly, the release of pro-oxidants induced by the destruction of the cell membrane during the freezing processes might lead to the oxidation and deterioration of pork patties [4].

At present, some novel technologies, including alternating electric field [5], infrared and microwave [6], composite films, and plant active extracts [7] are used to effectively maintain meat product quality. Some authors [8,9] found that extracts can lead to meat products having lower lightness and lipid oxidation compared to controls, thereby maintaining the quality of meat products during frozen storage. Further, fiber, as a water binder, improved water retention in reheated patties, delayed lipid oxidation in patties, and extended the shelf life of beef patties [10]. Recently, research has been increasingly drawn toward the study of mushrooms on meat production because their antioxidant compounds, including polyphenols, polysaccharides with antimicrobial and antioxidant activities, ergot thionine, and bacteriostatic enzymes can outstandingly fortify the whole physicochemical properties of meat products [11]. Moreover, meat products have obtained some typical flavors and nutrients from mushrooms [12].

*Pleurotus eryngii* (Pe) is widely cultivated in many Asian countries and is rich in nutrients, such as amino acids, proteins, minerals, fiber, and others. Furthermore, studies have reported that some superior bioactive molecules from it, such as polysaccharides, polyphenols, peptides, and sterols, exert a therapeutic impact on human diseases [13]. Therefore, there is a desire to form a balanced diet by adding Pe into processed meats [14], which would contribute to improving the nutrients and quality of meat products. For instance, some studies have reported the application of Pe to replace pork back fat [15], or have noticed the potential commercial applicability of Pe for processed meats to improve texture [16]. However, insufficient data are available concerning the impact of Pe on the quality changes of frozen pork patties. Hence, this essay explores the inhibitory effects of different concentrations of Pe (0, 0.25%, 0.50%, 1.00%, 2.00%) on the quality (pH, water holding capacity, color, texture, microstructure, and appearance property) deterioration and oxidation reactions (protein and lipid oxidation) of pork patties during freezing and thawing cycles (F–T cycles). By comparing these differences in properties, we can determine which level significantly improves the quality and frozen stability of pork patties.

## 2. Materials and Methods

### 2.1. Materials

Pork shoulder and neck, *Pleurotus eryngii* were purchased from Carrefour Supermarket (Harbin, Heilongjiang, China); Thiobarbital acid (TBA), guanidine hydrochloride, 2, 4-Dinitrophenylhydrazine (DNPH), trichloroacetic acid (TCA), Coomassie brilliant blue (CBB), and bovine serum albumin (BSA) were purchased from Shuanghuan Technology Development Co., Ltd. (Harbin, Heilongjiang, China). All other reagents used were analytical grade (Shuanghuan Technology Development Co., Ltd., Harbin, Heilongjiang, China).

### 2.2. Preparation of Pe Powder

Pe was washed with tap water, cut, and dried to a constant weight at 50 °C (DHG-9140A, Blast drying oven, Jinan Laibao Medical Equipment Co., Ltd., Jinan, China), then ground into powder by a Pulverizer (FW80, Tianjin Taist Instrument Co., Ltd., Tianjin, China). Finally, the Pe powder was sieved through a 100 mesh sieve to obtain the same particle size and stored in a dryer for use.

### 2.3. Preparation of Pork Patties

After being removed from the adipose tissue and fascia, the lean pork was cut into equal amounts and mixed with a silent cutter (JR05-300, Zhejiang Supor Co., Ltd., Taizhou, Zhejiang, China). The Pe powder (0.25%, 0.50%, 1.00%, and 2.00%) was added based on the meat weight, approximately 30 g per minced meat dough. After thoroughly mixing the meat and incorporated ingredients (we performed the same stirring operation on the blank group as the experimental group), we shaped the mixture in a mold (we made circular pork patties with a diameter of 6 cm and a thickness of 1 cm), and wrapped them in film. Each group has 3 parallel samples, totaling 15 samples. The first freezing-thawing treatment of the test sample was to store it in a −20 °C refrigerator for 7 days, then the samples were thawed in a 4 °C refrigerator for 12 h. The 0, 1, 3, and 5 F–T treatments were performed in sequence. After each treatment ended, a certain amount of sample was used for the determination of relevant indicators.

### 2.4. pH Value

Pork samples (5 g) and 45 mL of sterile deionized water were mixed and homogenized using an ultra-high-speed homogenizing emulsifier (Y25, Shanghai Yuedi Machinery Equipment Co., Ltd., Shanghai, China) for 60 s. After this, a precision pH meter (PHS-3E, Shanghai Yidian Scientific Instrument Co., Ltd., Shanghai, China) was used to determine the pH of the mixture [17].

### 2.5. Water Holding Capacity

#### 2.5.1. Centrifugal Loss

The pork samples were made to weigh 5 g before centrifuging [18]. After the samples were centrifuged at 9000 rpm for 10 min at 4 °C using a high-speed centrifuge (HL-16KS, Shanghai Chuangbo Biotechnology Co., Ltd., Shanghai, China), the water on the surface of the samples was removed with degreased cotton, and the samples were weighed again. This weight was recorded as m_0_. Centrifugal loss was calculated using the following formula:C/% = (5 − m_0_)/5 × 100,(1)
where m_0_ is the masses (g) of the sample after centrifuging.

#### 2.5.2. Thawing Loss

The thawing loss was calculated as a percentage of the weight loss of the pork samples under different F–T cycles. Briefly, the pork samples were weighed before freezing (m_1_). After the samples were thawed, the water on the surface of the samples was removed with paper towels, and the samples were weighed again. This weight was recorded as m. Thawing loss was calculated using the following formula:T/% = (m_1_ − m)/m × 100,(2)
where m_1_ and m are the masses (g) of the sample before freezing and after thawing, respectively.

### 2.6. Color and Texture

After thawing, the sample was placed in the air to react for 30 min, and was measured using a colorimeter (ZE-6000, Shanghai Shouli Industrial Co., Ltd., Shanghai, China). The equipment was calibrated with a whiteboard (L* = 90.26, a* = −1.29, b* = 5.18) to determine the L* value, a* value, and b* value of the pork patty.

Using a texture analyzer (Brookfield CT-3, Shanghai Weichuan Precision Instrument Co., Ltd., Shanghai, China), the sample was steamed and cut into sections with a height of 20 mm and a diameter of 8 mm. A P/50 probe was used, with a pre-test rate of 2.0 mm/s, a test rate of 2.0 mm/s, and a post-test rate of 2.0 mm/s. The measurement mode was “strain” 40%, and 3 parallel samples were measured in each group, taking the average value.

### 2.7. Lipid Oxidation and Protein Oxidation

#### 2.7.1. Lipid Oxidation

Peroxide value (POV) was determined with some modifications [14]. First, 3 g thawed minced meat was homogenized at high speed with 20 mL of chloroform-methanol mixed solution for 30 s. Next, 3 mL of 0.5% NaCl solution was added to the mixture and was centrifuged at 3000 rpm for 10 min. After 10 mL of the clear solution was taken and mixed with an equal volume of chloroform-methanol mixed solution in a glass test tube, 25 μL of ammonium thiocyanate solution and 25 μL of ferrous chloride solution were added successively, both vortexed for 3 s. After this, the glass test tube was placed at room temperature for 5 min, and its absorbance was measured at 500 nm using a Microplate reader (Epoch 2, Beijing Zhiyan Technology Co., Ltd., Beijing, China).

Lipid oxidation was evaluated by measuring the thiobarbituric acid reactive substances (TBARS) referring to [19] with minor alterations. In brief, 2 g of the sample was placed in a stoppered graduated test tube. Next, 3 mL of TBA solution and 17 mL of TCA solution were added into the tube. The reaction solution was in a boiling water bath for 30 min after being mixed well. Finally, the obtained solution was centrifuged for 10 min (3000 rpm/min) at 4 °C, and the absorbance was measured at 532 nm.

#### 2.7.2. Protein Oxidation

The degree of protein oxidation was analyzed by the method of protein sulfhydryl and carbonyl content determination [20]. Using BSA as the standard, the protein content was determined using the Coomassie brilliant blue method. A total of 2.5 mL Tris Gly 8M urea solution was added to 0.5 mL of sample homogenization solution, shaking thoroughly to mix duplicating two parts well. A measure of 20 μL Ellman reagent was added to the experimental group; 20 μL Tris Gly solution was added to the control group water bath at 37 °C for 15 min, it was centrifuged at 10,000 rpm at 4 °C for 5 min, and the absorbance value of the supernatant at 412 nm was determined. 

In brief, 3 g of samples and 27 mL of physiological saline were placed in a 50 mL centrifuge tube. After this, the mixture was homogenized at 10,000 rpm for 1 min. Next, 2 parts of 0.1 mL sample homogenization solution were respectively added into the following two groups: (1) 0.5 mL of 2 M HCl; (2) 0.5 mL of 2 M HCl (containing 0.02 M DNPH). After being heated at 37 °C for 15 min in a water bath, the two groups were added to 0.5 mL of 20% TCA solution and centrifuged at 10,000 rpm at 4 °C for 10 min to obtain the precipitate. The 2 parts of the precipitates were washed three times with a mixture of alcohol and ethyl acetate and were centrifuged again at 10,000 rpm at 4 °C for 10 min. Finally, the 2 parts of the precipitates were separated and both in 1 mL 6 M GuCl solution at 37 °C for 15 min, and the carbonyl content was determined spectrophotometrically by reading the absorbance of the solution at 370 nm; it was expressed using an extinction coefficient of 2.1 × 10^4^ L·M^−1^·cm^−1^.

### 2.8. SEM Determination

Slice (2 mm) × 2 mm × The 2.5 mm pork patty sample was immersed in a 2.5% glutaraldehyde solution for approximately 26 h and dehydrated with a series of ethanol solutions (60–100%). Then, isoamyl acetate was used instead of ethanol, and the sample was freeze-dried using an FD-ID-50 freeze-drying machine (Bilang Instrument Co., Ltd., Shanghai, China). After this, dry samples were sputtered and gold-plated using an ETD-3000 ion sputtering instrument (Beijing Jinggong Technology Development Co., Ltd., Beijing, China). Under an acceleration voltage of 3.0 kV, the microstructure was observed using a Carl Zeiss microscope (Oberkochen, Germany).

### 2.9. Statistical Analysis

The entire experimental design was followed in triplicate (*n* = 3), and each value was presented as the mean and standard deviation. The effect of different Pe treatments on pH value, water holding capacity, lipid and protein oxidation, color, and texture parameters were examined using a mixed-model ANOVA, where different Pe treatments served as the fixed effect and replicates served as the random effect. The pairwise differences between least-square means were evaluated using Duncan’s method. If *p* < 0.05, the differences were considered significant. Significant differences were analyzed using SPSS Statistics 24 software and plotted using SigmaPlot 12.5 software.

## 3. Results and Discussion

### 3.1. pH Value

The pH value is considerably important to the physical and chemical properties of meat; it involves water retention, gel property, elasticity, and water retention capacity [21]. And the production of acidic (thiols and organic acids, etc.) and alkaline compounds (ammonia and amines, etc.) during the food component degradation frequently leads to fluctuations in pH [22]. It can be seen from Figure 1 that the pH of pork patties as a whole fluctuated around 6.5 before F–T cycles [23]. At this time, during the ripening-autolysis period, glycogen decomposition leads to fat dissolution, free fatty acid and other acidic substances. In the process of minced meat products, protein was destroyed and free amino acids were generated [24]. In addition, because of the high content of basic amino acids in Pe powder, the pH of the fresh sample gradually approached 7 with the increase in the added amount.

The formation of ice crystals damaged the cell tissue of muscle and induced the large amount of juice loss, which reduced the pH value of pork patties reduced during F–T cycles [25]. In addition, the increasing in concentration of the exudate resulted in the salting out of meat during freezing, which caused protein denaturation and gradually reduced the soluble protein in the solution [26], breaking the acid-base balance of meat tissue. And, the concentration of hydrogen ions in the solution increased leading to the pH of pork patties decrease. Therefore, the pork patties at five F–T cycles were acidic between 6.13 and 6.37. Among all groups, only the pH of pork patties with 0.50% Pe showed an slightly decreasing trend, which may be related to Pe inhibiting the deterioration of F–T pork patties. And the pH value of pork patties with 0.50% Pe (6.37) is higher than the control (6.13) by 0.24 at five F–T cycles. It can also be seen that the addition of 0.25%, 1.00% and 2.00% Pe (6.25, 6.36, 6.35) can also alleviate the changes of pork patties on pH at five F–T cycles. This may be due to the abundance of basic amino acids and natural buffering capacity from Pe powder, or to the reduction of protein denaturation or the overall drip loss caused by salting out during freezing after addition. A research showed that the addition of winter mushroom powder to pork patties inhibited the growth and reproduction of putrefactive bacteria and improved the water holding capacity of meat production [11]. It was also suggested that frozen meat with higher pH would have better functional characteristics [27]. In conclusion, 0.50% Pe significantly maintain the pH property of pork patties during freezing and thawing cycles.

### 3.2. Water Holding Capacity

The pork patties’ water-holding capacity (WHC) was evaluated by the centrifugal loss and the thawing loss (Figure 2): the more water loss, the worse the water retention ability of muscles. With the increase in F–T cycle times, all samples’ thawing loss and centrifugal loss were increased. Among them, the inhibition degree is related to the additional amount of Pe. The greater the quantity of mushrooms added, the more enhanced the water-holding capacity of the patty was. The centrifugal loss increased to the highest level at five F–T cycles, 14.14%, 10.68%, 9.29%, 8.78%, and 7.63% for 0-2.00%, respectively (Figure 2a). The thawing loss of the control reached 12.55% after five F–T cycles, whereas the patties treated with Pe reached 9.59% (0.25%), 8.57% (0.50%), 6.33% (1.00%), and 3.14% (2.00%), respectively (Figure 2b). These results highlight the remarkable help of Pe in reducing the water loss in patties under F–T treatments. The formation of ice crystals can destroy the ultrastructure of muscle tissues, and the frozen muscle fibers exhibit limited capacity to absorb water molecules during prolonged thawing, resulting in significant water loss and diminished meat elasticity [28]. The study showed that dietary fiber in Pe had oil absorption and hydration properties [29], which aggravated the influence of dietary fiber on the gel network structure in meat products. Adding cellulose can increase water and fat retention in meat products [30,31], which greatly improves the WHC of pork patties. Moreover, other authors [32] found that Saccharides could inhibit the growth and recrystallization of ice through specific interactions with ice crystals. Based on the results, the increase in pH of the pork patties containing Pe simultaneously increased the net negative charge of the meat proteins [11], which interrupted the fat and water to flow out. Therefore, Pe improved the stability of WHC of pork patties due to polysaccharide in Pe electrostatic interaction with proteins [33], which enhanced protein structure stability.

### 3.3. Color Value

Color, as a desirable attribute for meat products, often intuitively influences the consumer’s choice [3]. The addition of Pe decreased the lightness (L* value) of the samples. The L* values of 1.00% and 2.00% Pe samples were significantly (*p* < 0.05) lower than the control (Table 1). The changes in L* values may be related to the addition of Pe used in the processing, which contained pigments and had a white appearance [34]. The increasing L* values of samples presented the decreased acceptability and freshness of pork patties. During freezing processes, the muscle fiber contraction induces water loss, which accumulates on the surface of the muscle, thus increasing the L* values of samples [35]. The positive impact of Pe powder on pork patties can be seen by the lower L* values of the experimental group compared to the control. Among them, the L* values of patties with 2.00% Pe are the lowest (*p* < 0.05), which may be related to the highest water retention ability of cellulose. With the increase in F–T cycle times, the redness (a* values) of all patties decreased due to the increasing red pigment destruction and destroyed myoglobin [36]. Some lipid oxidation products induce the oxidation of ferrous heme iron and stimulate the transformation from oxymyoglobin to metmyoglobin [37]. As shown in Table 1, the addition of Pe powder can decrease the reduction of the redness of the patty during F–T cycles. In particular, pork patties with 0.50% Pe had the lowest reduction of a* value after 5 F–T cycles, which may be related to the antioxidant capacity of the components in Pe. At the same time, the loss of juice from the patties was accompanied by the loss of pigment during F–T cycles; this was also a reason for the decrease in the a* value of samples. All samples’ yellowness (b* value) increased initially, then decreased, and then increased again during F–T cycles. The changes in the yellowness of pork patties during freezing were mainly caused by fat oxidation [38], and the free radicals produced by the oxidation of highly unsaturated fatty acids on the cell membrane may cause the reaction of amines in proteins and the production of yellow pigment [39]. The b* values of pork patties with 0.50% Pe are 0.96-fold higher than those of the control group after 5 F–T cycles. Therefore, the pork patties with 0.50% Pe had the highest a* value among samples, and their L* values and b* values were both lower than the control group after 5 F–T cycles, which suggested that 0.50% is the best addition.

### 3.4. Texture

The sensory acceptability of meat products for consumers can be predicted by the results of texture determination, which is a primary parameter in the evaluation of the quality of meat products. Five parameters are useful for detecting the changes in the textural characteristics of samples, including hardness(g), adhesion(g), cohesiveness, elasticity(mm), gumminess(g), and chewiness (mJ). Table 2 shows that the hardness of all pork patties decreased significantly (*p* < 0.05). The addition of dietary fiber may affect the gel of meat protein, thus reducing the strength of the gel and leading to a softening of the texture [40]. These trends were consistent with the experimental results of [41]. In addition, the formation of large ice crystals from extracellular damaged the cellular structure, tore muscle fibers, and caused the tissue to become soft and tender [42]. At five freezing and thawing times, the cohesiveness, elasticity, and chewiness of pork patties were increased, and the adhesion of all patties was decreased instead. Some authors [43] pointed out that muscle fibers become more concentrated and aggregated due to water loss, which leads to firmer tissue structures. Moreover, the oxidation and degradation of protein caused changes in the texture of pork patties [44]. In addition, the gumminess of all patties increased under 5 F–T cycles except for the 2.00% addition, which is associated with its lowest hardness values. It can be seen from the increased chewiness of all pork patties that 0.50% Pe treatment has the lowest degree. Above all, pork patties incorporated with 0.50% Pe had 538.00 g lower hardness, 73.50 g lower adhesion, 0.48 lower cohesiveness, 2.49 mm lower elasticity, 540.00 g lower gumminess, and 13.30 mJ lower chewiness after 5 F–T cycles, compared to the control. The addition of 0.50% Pe significantly reduced the change of pork patties at a higher level under 5 F–T cycles. These results demonstrated that 0.50% Pe had a positive effect on inhibiting the degradation of patty quality.

### 3.5. Appearance

Appearance of pork patties is a direct characteristic that importantly determines the purchase intention of consumers. The effects of Pe on the appearance of pork patties after F–T treatments are shown in Figure 3. The redness of pork patty in each group decreased significantly with the increase in F–T cycle times (*p* < 0.05), which was consistent with the trend of the measured color value. The phenomenon may be associated with fat reduction. Moreover, oxymyoglobin is converted to ferrimyoglobin under the influence of oxygen free radicals in meat, and the reduction of heme in oxymyoglobin causes the blood red to become lighter [3]. In the experimental group, the addition of Pe powder had a positive effect on the maintenance of meat color. Variances for appearance can be particularly found between the experimental group with 0.50% Pe and the control group after frozen storage, where the redness of the former is reduced to a lower degree after five F–T cycles. In addition, the outflow of water from muscle cells led to lighter redness during the freezing and thawing processes [45]. In contrast, the experimental group with 1.00% and 2.00% Pe had a deeper color appearance than the control at five F–T cycles. This result may be involved with the color of Pe powder and endogenous protease in Pe that could promote the oxidation of the sample [46]. The trend is consistent with them [20]. Moreover, some authors figured out that it is difficult for consumers to accept an excessive amount of mushroom powder [47]. Therefore, the frozen pork patties have a higher visual acceptance at 0.50% addition.

### 3.6. Lipid Oxidation

The degree of lipid oxidation in meat and meat products can be determined using peroxide value (POV) and thiobarbituric acid reactive substances (TBARS) values.

Peroxide substance is the primary decomposition product of fat rancidity, which can indicate the degree of fat oxidation. As shown in Figure 4a, the POV of all samples increased initially and then decreased with the increasing F–T cycles. Compared to the control, patties with 0.50% Pe had significantly lower POV simultaneously (*p* < 0.05). During F–T cycles, the mechanical injury induced by mixing the meat caused the production of reactive oxygen species (ROS) and promoted the Millard reaction and lipid peroxidation, thence inducing a non-enzymatic browning reaction [48,49]. Similarly, mushrooms added to pork patties led to pro-oxidation immediately, thus promoting the volatile compounds to burst [11]. The POV of all pork patties tended to display an upward trend with the increase in freezing-thawing cycles (*p* < 0.05). After five F–T cycles, the addition of 0.50% and 1.00% Pe significantly inhibited the lipid oxidation of pork patties (*p* < 0.05), and their POV was 0.87 and 0.91-fold that of the control group, respectively. Inhibition of lipid oxidation is due to the fact that Pe contains such natural antioxidant components as polyphenols, polysaccharides, lysergic acid, etc., which have the ability to reduce iron and metal chelation, reducing the excitation of metal ions and oxygen free radicals on lipid oxidation [50]. However, 1.00% Pe and 2.00% Pe significantly increased the POV of pork patties during F–T cycles compared with 0.50% Pe treatment, and this trend is consistent with their results [20], which were affected by abundant lipoxygenase and polyphenol oxygenase from Pe. Some authors found that mushrooms contain bacteriostatic protein, which will reduce the effect of microbial enzymes on the oxidation of patties [51]. Therefore, the effect of 1.00% and 2.00% addition is not good because the degree of promoting oxidation is greater than that of inhibiting oxidation.

Thiobarbituric acid reactive substances (TBARS) value reflects the amount of decomposition products of hydroperoxides in meat, which is the content of malondialdehyde (MDA), the secondary metabolite of lipid oxidation, helpful for viewing the whole process of lipid oxidation comprehensively. As presented in Figure 4b, the TBARS values of all pork patties tended to trend upwards with the increase in freezing and thawing cycles (*p* < 0.05). After five cycles, the addition of Pe decreased the yield from the secondary stage process of fat oxidation, and 0.25–2.00% Pe successively reduced by 0.06, 0.25, 0.18, and 0.08 mg MDA/kg significantly compared with the control (*p* < 0.05,). The experimental group with 0.50% Pe was 0.66-fold lower than the control group, presenting the lowest MDA content of all the samples. Moreover, some authors [52] revealed that the antioxidant components in mushroom powder could inhibit lipid oxidation through metal ion chelation, which is attributed to antioxidants from Pe, such as polyphenols and lysergic acid. In general, phenolic compounds could chelate metal ions and scavenge free radicals [53]; lysergic acid could scavenge hydroxyl radicals by inhibiting hydroxyl radicals formed by the Fenton reaction of hydrogen peroxide [54]. Furthermore, some studies have reported that the extracts, proteins, and hydrolysates of mushrooms could effectively inhibit lipid oxidation [55]. This natural property of Pe could effectively exert the function of anti-lipid oxidation in meat products. Combined with Figure 4, the influence of Pe on preventing the generation of MDA is the reason why peroxidants effectively decreased under 3–5 F–T cycles; however, the decreasing trend of control in Figure 4a is only due to the acceleration of producing MDA from peroxidants. In conclusion, 0.50% Pe has the best effect on lipid antioxidation during F–T cycles. A similar conclusion has been obtained from another study [56].

### 3.7. Protein Oxidation

Myofibrillar protein accounts for approximately 55–65% of total muscle protein. Because the sulfhydryl groups of cysteine exposed on the surface of protein is constantly oxidized, there are disulfide bonds and other oxidation products generating [57]. In the process of myofibrillar protein oxidation, the molecules are stimulated to produce large active carbonyl molecules, which further change the original protein spatial structure. These protein oxidation processes change the structure of myofibrin and have a largely negative impact on meat quality [58], such as texture characteristics, ultimately leading to meat deterioration.

The reduction of sulfhydryl content is a major indicator of protein oxidation in meat products. As depicted in Figure 5a, the content of sulfhydryl groups in the pork patties gradually decreased with the increase in F–T cycles. The content of sulfhydryl groups in the pork patties with the addition of 0.50% and 1.00% Pe was 1.13, 1.07-fold (*p* < 0.05) higher than that of the control at 5 F–T cycles, but the addition of 5% Pe would promote the conversion of sulfhydryl groups structure to disulfide bond in myofibril protein. Some studies have shown that the presence of rich polyphenol oxidase and lipoxygenase in mushrooms is an essential reason for their easy browning and being putrefied during storage [59]. Moreover, this can also explain why the more mushroom powder added to meat products exerted no better influence instead, consistent with authors [60]. The oxidation of myofibrillar is closely related to lipid oxidation: the lipid oxidation products will change the stability of myofibrillar oxidation and reduction and form adducts with myofibrils through covalent modification, thus promoting myofibrillar oxidation [61]. This influence factor is reflected in the experimental group with 0.50% addition, which not only inhibited lipid oxidation but also reduced the denaturation; simultaneously, the rate of myofibrillar supplying H_2_O_2_ to lipid oxidation also decreased, thereby reducing the lipid oxidation products. Therefore, adding 0.50% Pe greatly delayed the oxidation rate of protein in pork patties at five F–T cycles.

The increase in carbonyl content is another major indicator of protein oxidation in meat products, and the increase in carbonyl content in myofibrillar protein will alter its own structure, further exacerbating the deterioration of meat and meat products. The content of carbonyl (Figure 5b) in the pork patties continuously increased during F–T cycles. Under the stimulation of peroxidants and the attack of ·OH, the protein will be degraded to produce peptides, free amino acids, etc; at the same time, the side chains of amino acids in the protein or free amino acids will be oxidized, which induced carbonylation. The trend of increase in the carbonyl content could be slowed after adding Pe at 5 F–T cycles: among treatments, carbonyl content reached the highest level with 0.25% Pe (6.41 nmol/mg protein) followed by 2.00% Pe (5.57 nmol/mg protein), 1.00% Pe (4.21 nmol/mg protein), and 0.50% Pe (3.27 nmol/mg protein), whereas carbonyl content contributed to 6.63 nmol/mg protein in the control group at five F–T cycles. The experimental group with 0.50% Pe was 0.49-fold than the control group, presenting the lowest carbonyl content among all samples. The carbonyl group results of pork patties with 1.00% and 2.00% Pe is consistent with the trend of peroxide values determination (Figure 4) during F–T cycles. This reflects that lipid oxidation has the greatest impact on the formation of macromolecular structure and the spoilage process of patties, changing its characteristics and affecting its quality. Moreover, the protease in Pe will further hydrolyze the protein in the patties and produce many peptide chains after Pe was mixed evenly in patties. The side chain of its amino acid is easier to be oxidized, and carbonylation and disulfide bonding occur [62], which will accelerate the deterioration of the protein. In addition, the content of basic amino acids in patties treated with Pe exhibited increased [63], thereby the aminos among them are cross-linked with the active carbonyl groups generated from protein oxidation, to form Schiff base and H_2_O_2_ [64]. These are reasons for the addition of Pe powder more than 0.50% tending to prooxidation. Therefore, 0.50% Pe has the best antioxidant effect on protein of pork patties.

### 3.8. Microstructure

Scanning electron microscopy of meat and meat products reflects the arrangement of muscle fibers and their change process, which is crucial for this type of characterization, related to their texture and water retention [4]. As shown in Figure 6, the microstructure of the treated group with the addition of Pe was slightly denser compared to the control group, which may be related to the water absorption of the Pe powder [65]. After five freezing and thawing cycles (Figure 6), the microstructures of pork patties in each group gradually became hollow and more loosely shaped, with the control group having the most disordered microstructure with the generation of large aggregates accompanied by larger holes. The fluctuating temperature would cause the water in the pork patties to continuously generate and melt in the form of ice crystals, which caused more damage to the muscle cells and proteins, and the increase in the number of fluctuations would increase the degree of damage to the muscle and the volume of ice crystals formed, so that a rougher and more inhomogeneous structure was obtained, which further led to the decrease in water holding capacity and the deterioration of the texture. At the same time, the oxidative denaturation of muscle fibers will also lead to the original regular mesh structure into their own contraction of polymerized macromolecules, which will also lead to larger gaps in the patties, which introduces more oxygen and accelerates the deterioration of the meat. The addition of Pe can be used to maintain the stability of the intermuscle fiber structure in pork patties [66], but this stabilizing ability is related to its additive amount, and only the lowest degree of muscle deterioration and the best improvement effect was observed at the 0.50% additive in the micrographs after 5 F–T cycles. In addition, it can be seen from the graphs that the addition of Pe (especially more than 0.50%) significantly reduced the loss of water from the pork patties during freezing and thawing, which was mainly through the binding of myogenic fiber tissues, and at the same time altered the original tissue structure of the patties to obtain a softer texture, consistent with the trend of the results of other experiments in this paper. Pork patties in the 1.00–2.00% addition group after 5 F–T cycles remodeled the myofibrillar structure in the patties to a much greater extent than at the 0.50% addition due to the increased proportion of dietary fibers in the Pe, and they existed in a larger lamellar structure. It is suggested that excessive competition for water adsorption between fibers and proteins may also lead to disruption of the network structure [67]. It may also be related to the elevated proportion of some of these pro-oxidant components. Therefore, pork patties under 0.50% Pe treatment were beneficial in maintaining the structural integrity of their muscle fibers.

## 4. Conclusions

The addition of Pe to pork patties could improve WHC and decrease the degree of oxidation and color deterioration during the repeated F–T process compared to the control. Meanwhile, 0.50% Pe exhibited a more significant effect than others on inhibiting the quality and microstructure deterioration of pork patties. These suggest that Pe has a great potential for maintaining the quality stability of pork patties during frozen storage submitted to temperature fluctuations. In addition, the textural properties of patties were improved by the supplement of Pe. Our study showed that Pe has broad application prospects in improving the quality, stability, and shelf life of frozen raw comminute meat products.

## Figures and Tables

**Figure 1 foods-13-00501-f001:**
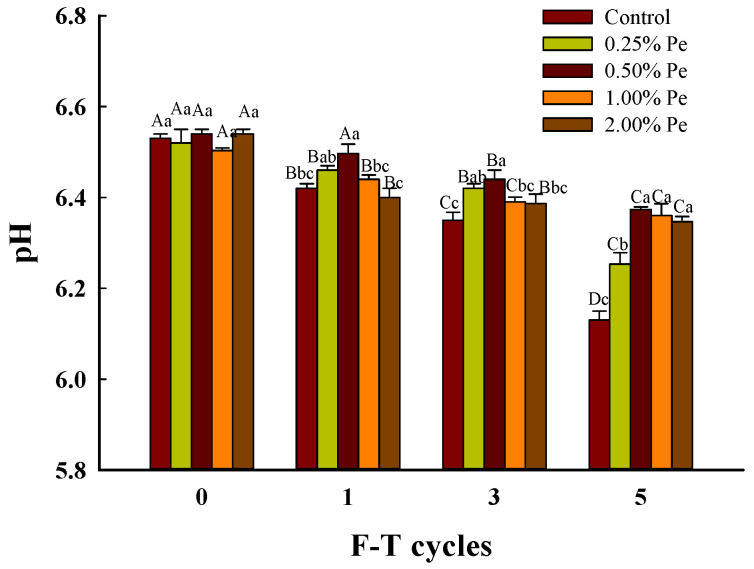
Effect of Pe powder on the pH value of pork patties during freezing and thawing cycles. Significant differences between the means for the same time of F–T cycles are denoted by different lowercase letters (a–c), and differences between the means for the same percentage of Pe powder are denoted by different uppercase letters (A–D; *p* < 0.05).

**Figure 2 foods-13-00501-f002:**
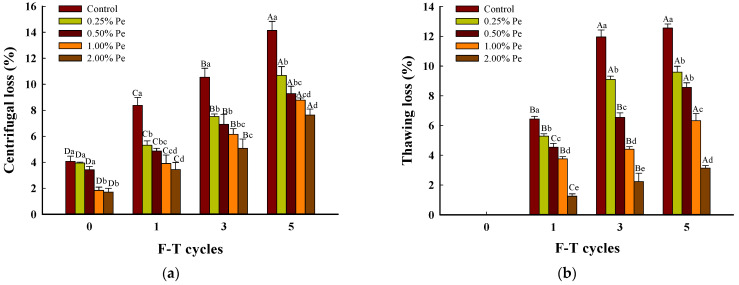
Effect of Pe powder on the Water loss of pork patties during freezing and thawing cycles ((**a**): Centrifugal loss, (**b**): Thawing loss). Significant differences between the means for the same time of F–T cycles are denoted by different lowercase letters (a–e), and differences between the means for the same percentage of Pe powder are denoted by different uppercase letters (A–D; *p* < 0.05).

**Figure 3 foods-13-00501-f003:**
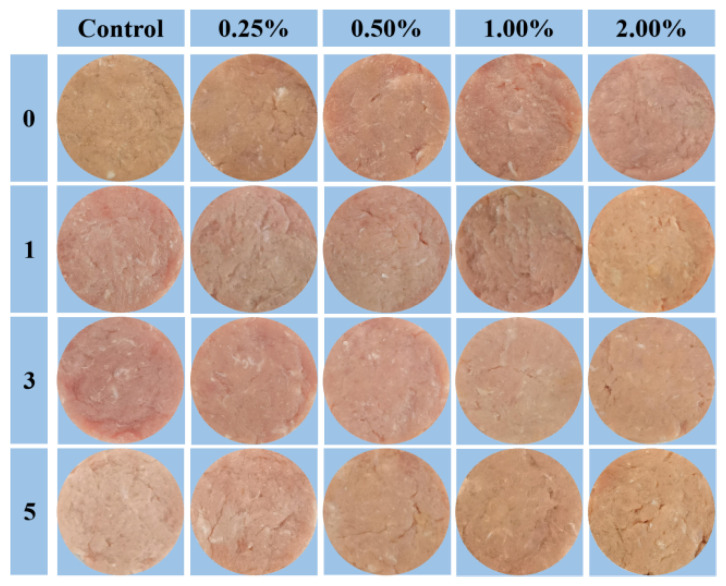
Effects of Pe powder on the appearance of pork patties during freezing and thawing cycles.

**Figure 4 foods-13-00501-f004:**
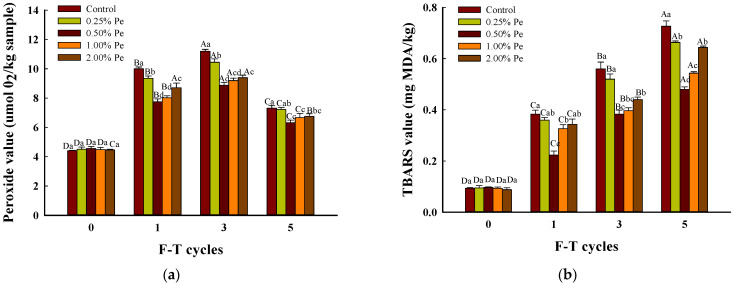
Effects of Pe powder on the lipid oxidation of pork patties during freezing and thawing cycles ((**a**): POV, (**b**): TBARS). Significant differences between the means for the same time of F–T cycles are denoted by different lowercase letters (a–d) and differences between the means for the same percentage of Pe powder are denoted by different uppercase letters (A–D; *p* < 0.05).

**Figure 5 foods-13-00501-f005:**
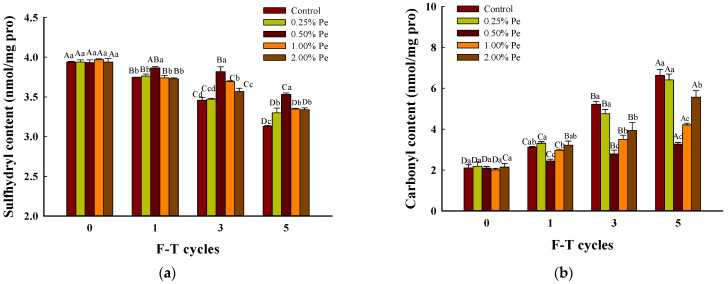
Effect of Pe powder on the protein oxidation of pork patties during freezing and thawing cycles ((**a**): Sulfhydryl, (**b**): Carbonyl content). Significant differences between the means for the same time of F–T cycles are denoted by different lowercase letters (a–d), and differences between the means for the same percentage of Pe powder are denoted by different uppercase letters (A–D; *p* < 0.05).

**Figure 6 foods-13-00501-f006:**
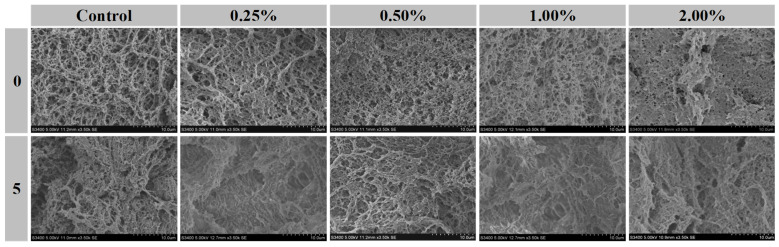
Effect of Pe powder on the microstructure (C, 3500×) of pork patties before and after 5 freezing and thawing cycles.

**Table 1 foods-13-00501-t001:** Effect of Pe powder on the color value of pork patties (*n* = 3) during freezing and thawing cycles.

Treatments			Color Value		
			L*	a*	b*
F–T cycles	0	Control	42.82 ± 0.78 ^Da^	10.39 ± 0.49 ^Aab^	15.28 ± 0.09 ^Bb^
		0.25%	42.71 ± 0.06 ^Ca^	9.46 ± 0.04 ^Ac^	14.33 ± 0.03 ^Bc^
		0.50%	41.75 ± 0.12 ^Dab^	10.61 ± 0.05 ^Aa^	14.24 ± 0.02 ^Cc^
		1.00%	41.00 ± 0.09 ^Cbc^	9.82 ± 0.09 ^Abc^	15.30 ± 0.07 ^ABb^
		2.00%	40.16 ± 0.09 ^Dc^	9.24 ± 0.06 ^Ac^	16.00 ± 0.23 ^Ba^
	1	Control	45.04 ± 0.35 ^Ca^	8.45 ± 0.01 ^Ba^	15.61 ± 0.1 ^Ac^
		0.25%	43.81 ± 0.02 ^Bb^	8.91 ± 0.42 ^ABa^	15.30 ± 0.07 ^Ac^
		0.50%	42.36 ± 0.07 ^Cc^	8.81 ± 0.39 ^Ba^	17.92 ± 0.27 ^Aa^
		1.00%	42.35 ± 0.17 ^Bc^	8.91 ± 0.09 ^Ba^	16.32 ± 0.15 ^Ab^
		2.00%	41.87 ± 0.14 ^Cc^	9.10 ± 0.13 ^Aa^	17.71 ± 0.06 ^Aa^
	3	Control	46.35 ± 0.18 ^Ba^	7.69 ± 0.65 ^BCb^	13.68 ± 0.07 ^Cb^
		0.25%	44.74 ± 0.08 ^Bb^	8.61 ± 0.08 ^Bab^	13.89 ± 0.10 ^Bab^
		0.50%	43.46 ± 0.11 ^Bc^	8.66 ± 0.23 ^Bab^	12.59 ± 0.36 ^Dc^
		1.00%	43.14 ± 0.05 ^Bcd^	8.82 ± 0.05 ^Ba^	13.83 ± 0.18 ^Cb^
		2.00%	42.80 ± 0.18 ^Bd^	8.54 ± 0.11 ^Bab^	14.37 ± 0.13 ^Ca^
	5	Control	54.13 ± 0.07 ^Aa^	6.95 ± 0.03 ^Cab^	15.62 ± 0.11 ^Aa^
		0.25%	53.68 ± 0.83 ^Aab^	6.98 ± 0.1 ^Cab^	14.51 ± 0.41 ^Bb^
		0.50%	51.96 ± 0.33 ^Abc^	7.25 ± 0.16 ^Ca^	15.06 ± 0.29 ^Bab^
		1.00%	50.19 ± 0.89 ^Acd^	7.12 ± 0.33 ^Cab^	14.49 ± 0.66 ^BCb^
		2.00%	48.50 ± 0.66 ^Ad^	6.68 ± 0.69 ^Cb^	15.84 ± 0.07 ^Ba^

Significant differences between the means of the same time of F–T cycles are denoted by different lowercase letters (a–d) and between the means for the same percentage of Pe powder are denoted by different uppercase letters (A–D; *p* < 0.05).

**Table 2 foods-13-00501-t002:** Effect of Pe powder on the texture of pork patties (*n* = 3) during freezing and thawing cycles.

Treatments			Hardness (g)	Adhesion (g)	Cohesiveness	Elasticity (mm)	Gumminess (g)	Chewiness (mJ)
F–T (cycles)	0	Control	2721.00 ± 275.15 Aa	199.00 ± 0.82 Aa	0.22 ± 0.04 Bb	3.02 ± 0.45 Bab	604.00 ± 49.56 Ba	17.90 ± 2.78 Ca
		0.25%	2361.33 ± 170.87 Aab	256.33 ± 58.90 Aa	0.21 ± 0.02 Cb	2.48 ± 0.06 Bb	520.33 ± 2.62 Ba	12.60 ± 0.43 Ba
		0.50%	2121.00 ± 168.12 Abc	251.67 ± 97.11 Aa	0.25 ± 0.02 Bb	2.58 ± 0.31 ABab	503.00 ± 26.99 Aa	12.90 ± 2.16 Aa
		1.00%	2116.67 ± 7.59 Abc	196.33 ± 3.77 Aa	0.34 ± 0.16 Aab	3.56 ± 1.28 Aab	638.33 ± 341.93 Aa	28.97 ± 21.20 Aa
		2.00%	1917.33 ± 109.14 Ac	279.67 ± 111.96 Aa	0.50 ± 0.12 ABa	4.86 ± 1.33 Aa	939.33 ± 190.93 Aa	46.90 ± 22.43 Aa
	1	Control	2578.33 ± 270.37 Aa	198.33 ± 0.47 Aa	0.46 ± 0.04 ABa	3.20 ± 0.09 Ba	698.50 ± 29.50 Ba	21.90 ± 0.30 BCa
		0.25%	2072.00 ± 130.26 Aab	182.67 ± 20.29 Aa	0.46 ± 0.04 Ba	2.93 ± 0.26 ABab	496.00 ± 36.00 Bb	13.60 ± 1.80 Bb
		0.50%	1947.00 ± 74.65 Ab	198.33 ± 0.47 ABa	0.27 ± 0.05 Bb	2.98 ± 0.11 Aab	484.50 ± 39.50 Ab	14.15 ± 1.65 Ab
		1.00%	1810.33 ± 270.04 Ab	192.00 ± 9.93 Aa	0.41 ± 0.10 Aab	2.72 ± 0.16 Ab	506.00 ± 6.00 Ab	13.15 ± 0.65 Ab
		2.00%	1824.67 ± 126.15 Ab	197.00 ± 2.16 ABa	0.41 ± 0.08 ABab	2.62 ± 0.09 Bb	501.50 ± 51.50 BCb	12.60 ± 1.40 Bb
	3	Control	1609.50 ± 492.50 Ba	129.50 ± 34.50 Aa	0.46 ± 0.03 ABab	4.07 ± 0.16 ABa	1103.00 ± 4.50 Aa	44.05 ± 1.95 ABa
		0.25%	969.00 ± 199.00 Bb	48.00 ± 5.00 Bbc	0.56 ± 0.05 Aa	3.04 ± 0.05 ABb	525.95 ± 85.00 Bb	15.60 ± 2.30 Bb
		0.50%	978.00 ± 172.50 Bb	96.00 ± 28.00 Bab	0.51 ± 0.05 Aa	3.00 ± 0.05 Ab	444.0 ± 28.00 Abc	13.05 ± 1.05 Abc
		1.00%	977.50 ± 126.50 Bb	77.50 ± 9.50 Cabc	0.50 ± 0.03 Aa	2.91 ± 0.07 Ab	487.00 ± 88.0 Abc	13.85 ± 2.15 Abc
		2.00%	603.50 ± 7.50 Bc	36.00 ± 14.00 Cc	0.34 ± 0.07 Bb	2.86 ± 0.20 Bb	359.50 ± 15.50 Cc	10.05 ± 0.25 Bc
	5	Control	1549.5 ± 25.5 Ba	133.00 ± 79.00 Aa	0.76 ± 0.24 Aa	4.61 ± 0.78 Aa	1198.00 ± 205.36 Aa	55.63 ± 17.04 Aa
		0.25%	941.50 ± 67.50 Bb	51.00 ± 14.00 Ba	0.55 ± 0.03 ABa	3.66 ± 0.56 Aab	958.33 ± 90.03 Aab	34.67 ± 7.88 Aab
		0.50%	538.00 ± 114.00 Cbc	73.50 ± 20.50 Ba	0.48 ± 0.07 Aa	2.49 ± 0.04 Bb	540.00 ± 50.84 Ab	13.30 ± 1.02 Ab
		1.00%	519.50 ± 2.50 Cbc	102.50 ± 7.50 Ba	0.59 ± 0.01 Aa	3.15 ± 0.66 Ab	751.33 ± 240.80 Ab	24.73 ± 13.22 Ab
		2.00%	277.00 ± 92.00 Cc	80.00 ± 4.00 BCa	0.60 ± 0.04 Aa	3.91 ± 0.26 ABab	742.00 ± 103.04 ABb	28.23 ± 2.66 ABab

Significant differences between the means for the same time of F–T cycles are denoted by different lowercase letters (a–c), and differences between the means for the same percentage of Pe powder are denoted by different uppercase letters (A–C; *p* < 0.05).

## Data Availability

Data is contained within the article.

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
