# Peer review of "Effect of Pleurotus eryngii on the Characteristics of Pork Patties during Freezing and Thawing Cycles"

_foods, 2024, doi:10.3390/foods13030501_

Round 1
Reviewer 1 Report
Comments and Suggestions for Authors
The research work entitled: "Effect of Pleaurotus Eryngii on the characteristics of pork patties during freezing and thawing cycles" aims to investigate the effect of the addition of the mushroom on the characteristics of pork patties during freezing cycles. freezing and thawing.
Pleaurotus Eryngii: without the a, Pleurotus.
Generally, I found the work not very interesting, I also don't understand why the authors considered these freezing and thawing cycles. The food industry is regulated regarding the storage and transport of food so freezing and thawing does not seem like correct practice to me. I report some observations that can be extended to the entire manuscript.
Introduction
It should be better structured and the reason for the different freezing and thawing cycles used should be well focused. Furthermore, the topics covered should be better linked to each other in order to facilitate reading.
Materials
The authors reported using 30g of minced meat to produce pork patties with measurements of 6cm per 1cm thick, which doesn't seem like much to me.
The authors did not report a chemical composition of the mushroom flesh and powder used for the study.
Methods
Review some mistakes.
Results
Please enter the entry: Results and discussions
In general, the discussion must be summarized for all the points covered but above all the results obtained must be highlighted. The proposed discussion concerns more what is found in the literature that does not support the results obtained by the authors themselves.
Author Response
Response to Reviewer #1
Q1: Generally, I found the work not very interesting, I also don't understand why the authors considered these freezing and thawing cycles. The food industry is regulated regarding the storage and transport of food so freezing and thawing does not seem like correct practice to me.
A1: Thanks for your suggestion, and we are delighted to share our views with you. Freeze-thaw cycles caused by temperature variation during freezing and frozen storage remain a major concern for processors and consumers. Meanwhile, repeated F-T cycles commonly occur in the world during long-distance transport, mainly caused by multiple transfers and very poor cold-chain conditions. And it is inevitable that temperature changes or repeated F-T cycles will occur in retail stores, restaurants or families. For example, the pork patties were thawed at a fast food restaurant, but they weren’t used up on the same day and were frozen again, presenting our research is necessary.
Thus, this manuscript shows the effect of F-T cycle on the pork patties, to investigate the influence of F-T cycle is important in terms of food supply. And the knowledge about F-T cycle causing quality properties damage for patties in this manuscript is useful for meat products industry.
The relative references are listed as follow:
Kaale, L. D., & Eikevik, T. M. (2014). The development of ice crystals in food products during the superchilling process and following storage, a review. Trends in Food Science & Technology, 39(2), 91–103.
Chen, Q. M., Xie, Y. F., Xi, J. Z., Guo, Y. H., Qian, H., Cheng, Y., Yao, W. R. (2018a). Characterization of lipid oxidation process of beef during repeated freeze-thaw by electron spin resonance technology and Raman spectroscopy. Food Chemistry, 243, 58–64.
Fu, Y., Cao, Y., Chang, Z. Y., Zou, C. J., Jiang, D. M., Gao, H. L., Jia, C. F. (2024). Effects of Flammulina velutipes polysaccharide with ice recrystallization inhibition activity on the quality of beef patties during freeze-thaw cycles: An emphasis on water status and distribution. Meat science, 209, 109420.
Lan, W. Q., Hu, X. Y., Sun, X. H., Zhang, X., & Xie, J. (2020). Effect of the number of freeze-thaw cycles number on the quality of pacific white shrimp (Litopenaeus vannamei): An emphasis on moisture migration and microstructure by LF-NMR and SEM. Aquaculture and Fisheries, 5(4), 193–200.
Ali, S., Zhang, W., Rajput, N., Khan, M. A., Li, C. B., & Zhou, G. H. (2015). Effect of multiple freeze-thaw cycles on the quality of chicken breast meat. Food Chemistry, 173, 808–814.
Pan, N., Dong, C., Du, X., Kong B.H., Xia X.F. Effect of freeze-thaw cycles on the quality of quick-frozen pork patty with different fat content by consumer assessment and instrument-based detection.Meat Science, 2020, 172(5):108313.
Cheng, H., Song, S., Jung, E., Jeong, J., Joo, S., & Kim, G. (2020). Comparison of beef quality influenced by freeze-thawing among different beef cuts having different muscle fiber characteristics. Meat Science, 169, 108206.
Q2: Introduction
It should be better structured and the reason for the different freezing and thawing cycles used should be well focused. Furthermore, the topics covered should be better linked to each other in order to facilitate reading.
A2: Thanks for your suggestion, we have listed some reasons for freezing and thawing cycles and relevant research references, and the structure and content of Introduction has been better sorted out for articulating the research of freezing and thawing cycles on meat products in line 30-56, as follow:
Pork patty is an important material in the production of fast food, like hamburger. In order to preserve the essential nutrients and prolong its shelf life, frozen storage plays a critical role to fight against lipid oxidation and protein oxidation of patties. However, the degradation degree of meat quality will increase because of oxidation and microbial growth during frozen storage and processing of patties. During the frozen storage process, it is common to repetitively thaw and freeze pork patties for producers, sellers and consumers [1]. The incomplete cold chain transportation systems or changes in transportation environments exacerbate the quality deterioration of pork patties, such as excessive water loss, darkening of meat color, deterioration of flavor, and decrease in tenderness [2]. There are two main reasons for pork patties tending to be spoiled. Firstly, the formation and recrystallization of ice crystals caused the destruction of cell tissue and the water mobility and redistribution, which led to the worse quality of patty [3]. Secondly, the release of pro-oxidants induced by destruction of the cell membrane during freezing processes might lead to the oxidation and deterioration of pork patties [4].
At present, some novel technologies, including alternating electric field [5], infrared and microwave [6], composite films and plant active extracts [7] were used to maintain meat products quality effectively. Some authors [8-9] found that extracts can lead to meat products have lower lightness and lipid oxidation compared with control, thereby maintaining the quality of meat products during frozen storage. Besides, fiber as a water binders, improved water retention in reheated patties, delayed lipid oxidation in patties and extended shelf life of beef patties [10]. Recently, research has been increasingly drawn toward the study of mushroom on meat production, because its antioxidant compounds, including polyphenols, polysaccharide with antimicrobial and antioxidant activities, ergot thionine and bacteriostatic enzyme can outstandingly fortify the whole physicochemical properties of meat products [11]. Moreover, meat products obtained some typical flavors and nutrients from mushroom [12].
Q3: Materials
The authors reported using 30g of minced meat to produce pork patties with measurements of 6cm per 1cm thick, which doesn't seem like much to me.
A3: Thanks for your suggestion and we fully agree with your opinion. We also considered this before our research, but this article tended to simulate the environment of frozen pork patties during cold chain transportation and procurement for consumption, in the laboratory on a small scale, so a 30g sample was selected for studying the effect of Pe on the storage stability of pork patties to reduce unnecessary waste. For actual production and application, the proportion can be further expanded, hoping you can understand.
Q4: Materials
The authors did not report a chemical composition of the mushroom flesh and powder used for the study.
A4: Thanks for your suggestion and we fully agree with your opinion. We have detected the 5.21% polysaccharide content in Pe before our research, and the result of 83.10% DPPH radical clearance rate, 82.81% ·OH radical clearance rate, 87.10% ABTS radical clearance rate from extracted polysaccharide. These studies inspired us to use Pe powder in the preservation of freezing pork patties. In the article, we have added analysis of the effective ingredients of Pe based on references, further analyzing the reasons for the improvement effect of treatment groups. Besides, the exploration of the mechanism of the effect of active ingredients of Pe on myofibrillar proteins is another part of our research content, so it was not included in this article, hoping you can understand.
Q5: Methods
Review some mistakes.
A5: Thanks for your suggestion. Changes are marked in red in the manuscript.
Q6: Results
Please enter the entry: Results and discussions
In general, the discussion must be summarized for all the points covered but above all the results obtained must be highlighted. The proposed discussion concerns more what is found in the literature that does not support the results obtained by the authors themselves.
A6: Thanks for your suggestion. We have made adjustments to the content of the article according to your suggestion. If our comprehension here is not accurate enough, we hope you can understand and welcome your specific guidance again, special thanks to you. There are mainly about summarized points, the highlighted results and discussion concerned more the results, as follow:
line 190: 3. Results and discussion
3.1. pH value
pH value is considerably important to the physical and chemical properties of meat, involving water retention, gel property, elasticity and water retention capacity [21]. And the production of acidic (thiols and organic acids, etc.) and alkaline compounds (ammonia and amines, etc.) during the food component degradation frequently lead to fluctuations in pH [22]. It can be seen from the Figure 1 that the pH of pork patties as a whole fluctuated around 6.5 before F-T cycles [23]. At this time, during the ripening-autolysis period, glycogen decomposition leads to fat dissolution, free fatty acid and other acidic substances. In the process of minced meat products, protein was destroyed and free amino acids were generated [24]. In addition, because of the high content of basic amino acids in Pe powder, the pH of the fresh sample gradually approached 7 with the increase of the added amount.
The formation of ice crystals damaged the cell tissue of muscle and induced the large amount of juice loss, which reduced the pH value of pork patties reduced during F-T cycles [25]. In addition, the increasing in concentration of the exudate resulted in the salting out of meat during freezing, which caused protein denaturation and gradually reduced the soluble protein in the solution [26], breaking the acid-base balance of meat tissue. And, the concentration of hydrogen ions in the solution increased leading to the pH of pork patties decrease. Therefore, the pork patties at five F-T cycles were acidic between 6.13 and 6.37. Among all groups, only the pH of pork patties with 0.50% Pe showed an slightly decreasing trend, which may be related to Pe inhibiting the deterioration of F-T pork patties. And the pH value of pork patties with 0.50% Pe (6.37) is higher than the control (6.13) by 0.24 at five F-T cycles. It can also be seen that the addition of 0.25%, 1.00% and 2.00% Pe (6.25, 6.36, 6.35) can also alleviate the changes of pork patties on pH at five F-T cycles. This may be due to the abundance of basic amino acids and natural buffering capacity from Pe powder, or to the reduction of protein denaturation or the overall drip loss caused by salting out during freezing after addition. A research showed that the addition of winter mushroom powder to pork patties inhibited the growth and reproduction of putrefactive bacteria and improved the water holding capacity of meat production [11]. It was also suggested that frozen meat with higher pH would have better functional characteristics [27]. In conclusion, 0.50% Pe significantly maintain the pH property of pork patties during freezing and thawing cycles.
3.2. Water holding capacity
The Water holding capacity (WHC) of the pork patties was evaluated by the centrifugal loss and the thawing loss (Figure 2). The more water loss, the worse water retention ability of muscles. With the increase of F-T cycle times, the thawing loss and the centrifugal loss of all samples were increased. And among them, the inhibition degree is related to the addition amount of Pe. The greater the quantity of mushroom added, the enhanced water holding capacity of the Patty. The centrifugal loss increased to the highest at five F-T cycles, being 14.14%, 10.68%, 9.29%, 8.78%, 7.63% for 0-2.00% respectively (Figure 2a). The thawing loss of the control reached 12.55% after five F-T cycles, whereas the patties treated with Pe reached 9.59% (0.25%), 8.57% (0.50%), 6.33% (1.00%), 3.14% (2.00%) respectively (Figure 2b). These results suggested the remarkable help of Pe to reduce the water loss in patties under F-T treatments. The formation of ice crystals can destroy the ultrastructure of muscle tissues, and the frozen muscle fibers exhibited limited capacity to absorb water molecules during prolonged thawing, resulting in significant water loss and diminished meat elasticity [28]. The study showed that dietary fiber in Pe had properties of oil absorption and hydration [29], which aggravated to the influence of dietary fiber on the gel network structure in meat products. Adding cellulose can increase water and fat retention in meat products [30-31], which greatly improve the WHC of pork patties. Moreover, other authors [32] found that Saccharides could inhibit the growth and recrystallization of ice through specific interactions with ice crystals. Based on the results, the increase in pH of the pork patties containing Pe simultaneously increased net negative charge of the meat proteins [11], which interrupted the fat and water to flow out. Therefore, Pe improved the stability of WHC of pork patties due to polysaccharide in Pe electrostatic interaction with proteins [33], which enhanced the stability of protein structure.
3.3. Color value
Color, as an desirable attribute for meat products, often intuitively influences the consumer's choice [3]. The addition of Pe decreased the lightness (L* value) of the samples. The L* values of 1.00% and 2.00% Pe samples were significantly (P < 0.05) lower than the control (Table 1). The changes in L* values may be related to the addition of Pe used in the processing, which contained pigments and had a white appearance [34]. The increasing L* values of samples presented the decreased acceptability and freshness of pork patties. During freezing processes, the muscle fiber contraction induced the water loss, which accumulated on the surface of the muscle. Thus increasing the L* values of samples [35]. The positive impact of Pe powder on pork patties can be seen from the lower L* values of experimental group, compared with the control. Among them the L* values of patties with 2.00% Pe is the lowest (P < 0.05), which may related to the highest water retention ability of cellulose. With the increase of F-T cycles times, the redness (a* values) of all patties were decreased due to the increasing red pigment destruction and destroyed myoglobin [36]. Some lipid oxidation products induce the oxidation of ferrous heme iron and stimulate the transformation from oxymyoglobin to metmyoglobin [37]. As shown in Table 1, the addition of Pe powder can decrease the reduction of the redness of patty during F-T cycles. In particular, pork patties with 0.50% Pe had the lowest reduction of a* value after 5 F-T cycles, which may be related to the antioxidant capacity of the components in Pe. At the same time, the loss of juice in patties was accompanied by the loss of pigment during F-T cycles, which was also a reason of the decrease in a* value of samples. The yellowness (b* value) of samples were all increased firstly, decrease secondly, and increased finally during F-T cycles. The changes in the yellowness of pork patties during freezing was mainly caused by fat oxidation [38], and the free radicals produced by the oxidation of highly unsaturated fatty acids on the cell membrane may cause the reaction of amines in proteins and the production of yellow pigment [39]. The b* values of pork patties with 0.50% Pe are 0.96-fold than those of the control group after 5 F-T cycles. Therefore, The pork patties with 0.50% Pe had the highest a* value among samples, and its L* values and b* values are both lower than the control group after 5 F-T cycles, which suggested 0.50% is the best addition.
3.6 Lipid oxidation
In conclusion, the 0.50% Pe has the best effect of lipid antioxidation during F-T cycles. A similar conclusion have been obtained from other study [56].
3.7 Protein oxidation
These are reasons for the addition of Pe powder more than 0.50% tending to prooxidation. Therefore, 0.50% Pe has the best antioxidant effect on protein of pork patties.

Reviewer 2 Report
Comments and Suggestions for Authors
The revised manuscript is interesting and novel. It is important that the authors consider the following observations:
Line 2: rewrite… Pleurotus
Line 2: use italic text format for scientific names
Line 2: Use eryngii instead of Eryngii
Line 10: use italic text format for scientific names
Line 18: What´s POVE meaning?
Line 18: What´s TBARS meaning?
Line 15-22: Regarding the pH values, color and appearance, what results were obtained?
Line 41: Avoid using names or surnames of authors in the text; only the corresponding reference number should be assigned and placed in square brackets. Correct through the document.
Line 42: use italic text format for scientific names
Line 49: shelf life or shelf-life like in line 28. Homogenize through the document
Line 51: use italic text format for scientific names, correct through the manuscript
Line 51: did you mean period?
Line 66-61: Was the appearance evaluation omitted?
Line 74-78: include the suppliers of the reagents used
Line 80: Where was Pe obtained from? did you grow them? Did they acquire them commercially? were they donated?
Line 85: What was the level of salt and fat added to the burger?
Line 85: How are hamburgers stored? in polystyrene trays? Did they wrap themselves in film?
Line 95: use 12 h instead of 12 hours
Line 100: Was any equipment used to homogenize the samples?
Line 104: 5 g
Line 105: 9000 rpm for 10 min
Line 105: insert temperature of centrifugation process, as well as the equipment information (model, trademark, country)
Line 118: 30 min
Line 122-125: include equipment information for texture analyzer
Line 129: 3 g
Line 130: 30 s
Line 132: 10 min
Line 135: 3 s
Line 136: 5 min
Line 136: include equipment information for spectrophotomer
Line 139: did you mean…17 mL of TCA solution were…
Line 141: 30 min
Line 142: 10 min
Line 142: did you mean 3000 rpm?
Line 146: use the abbreviated form for bovine serum albumin, which was used at the beginning of this section. Note: it is recommended to only use the abbreviation after being mentioned in the text
Line 150: 15 min
Line 151: 5 min
Line 151: include temperature of centrifugation process
Line 153: 3 g
Line 154: 10,000 rpm
Line 154: 1 min
Line 155: 0.1 mL
Author Response
Response to Reviewer #2
Thank you very much for your recognition and comments on the paper. You are right that much revision and clarification need to be done. In addition, the details in the introduction, methodology, results and discussions process have been supplemented to reach the level of the journal. Those comments are all valuable and very helpful for revising and improving our paper, as well as the important guiding significance to our researches. We sincerely hope that you can give me the chance to publish the paper. The responds to the reviewers’ comments and all corrections in the paper are as follow:
Q1: Line 2- Line 42: rewrite… Pleurotus; use italic text format for scientific names; Use eryngii instead of Eryngii; use italic text format for scientific names; What´s POVE meaning?; What´s TBARS meaning?; Regarding the pH values, color and appearance, what results were obtained?
A1: Thanks for your review carefully. You are right. In the revised manuscript, the content you proposed have been corrected as follow:
(1)The sentences in line 2-42 have been corrected in line 1-28 as follow:
Effect of Pleurotus eryngii on the Characteristics of Pork Patties during Freezing and Thawing Cycles
Abstract: Temperature fluctuations severely damage the quality, oxidation stability and structure of pork patties. This study investigated the potential reasons of Pleurotus eryngii (Pe) to protect frozen pork patties from quality degradation caused by temperature fluctuations and promoted the application of a natural ingredient. In this experiment, the pH, the water holding capacity (WHC), the properties of color and texture, the appearance, the degree of protein and lipid oxidation and the microstructure of patties with different addition of Pe (0%, 0.25%, 0.50%, 1.00%, and 2.00%) were intensified during freezing and thawing (F-T) cycles. The results showed that patties with 0.50% Pe exhibited a distinguishable improvement in the changes of pH, WHC, the color and texture during F-T cycles (P < 0.05). With the times of F-T cycles increasing, 0.50% Pe was able to inhibit lipid oxidation of patties by decreasing peroxide value (POV) and thiobarbituric acid reactive substances (TBARS) value to 0.87, 0.66-fold respectively those of control group, and suppress protein oxidation of patties with a protein sulfhydryl content increasing to 1.13-fold and a carbonyl content decreasing to 0.49-fold than those of patties from control group (P < 0.05), after 5 F-T cycles. In addition, the figures of appearance and microstructure of samples indicated that 0.50% Pe effectively restrained deterioration of structure features from patties after 5 F-T cycles. Thus, the addition of Pe effectively maintained the characteristics of pork patties under F-T cycles by studying.
Keywords: pork patties; Freezing-Thawing cycles; characteristics; Pleurotus eryngii
(2)The sentences in line 2-42 have been corrected in lines to avoid using names or surnames of authors in the text, as follow:
Line 218-220: A research showed that the addition of winter mushroom powder to pork patties inhibited the growth and reproduction of putrefactive bacteria and improved the water holding capacity of meat production [11].
Line 242-248: The study showed that dietary fiber in Pe had properties of oil absorption and hydration [29], which aggravated to the influence of dietary fiber on the gel network structure in meat products. Adding cellulose can increase water and fat retention in meat products [30-31], which greatly improve the WHC of pork patties. Moreover, other authors [32] found that Saccharides could inhibit the growth and recrystallization of ice through specific interactions with ice crystals.
Line 304-306: Some authors [43] pointed that muscle fibers become more concentrated and aggregated duo to the water lost, which led to firmer tissue structures.
Line 338: The trend is consistent with them [20].
Line 379-380: Besides, some authors [52] revealed that the antioxidant components in mushrooms powder could inhibit lipid oxidation through metal ion chelation, which attributed to antioxidants from Pe, such as polyphenols and lysergic acid.
Line 392: A similar conclusion have been obtained from other study [56].
Line 414-415: Moreover, this can also explain why the more mushrooms powder added to meat products exerted not better influence instead, consistant with authors [60].
Q2: Line 49- Line 66: shelf life or shelf-life like in line 28. Homogenize through the document; use italic text format for scientific names, correct through the manuscript; did you mean period?; Was the appearance evaluation omitted.
A2: Thanks for your review carefully. You are right. In the revised manuscript, the content you proposed have been corrected as follow:
(1)The sentences in line 49-51 have been corrected in line 30-32 as follow:
Pork patty is an important material in the production of fast food, like hamburger. In order to preserve the essential nutrients and prolong its shelf life, frozen storage plays a critical role to fight against lipid oxidation and protein oxidation of patties.
(2)The sentences in line 61-66 have been corrected in line 66-72 as follow:
Hence, this essay explore the inhibit impacts of different concentration of Pe (0, 0.25%, 0.50%, 1.00%, 2.00%) on the quality (pH, water holding capacity, color, texture, microstructure and appearance property) deterioration and oxidation reactions (protein and lipid oxidation) of pork patties during freezing and thawing cycles (F-T cycles). Through comparing these differences in properties, determine which level significantly improves the quality and frozen stability of pork patties.
Q3: Line 74- Line 80: include the suppliers of the reagents used; Where was Pe obtained from? did you grow them? Did they acquire them commercially? were they donated?
A3: Thanks for your review carefully. You are right. In the revised manuscript, the content you proposed have been corrected in line 73-80 as follow:
2.1. Materials
Pork shoulder and neck, Pleurotus eryngii were purchased from Carrefour Supermarket (Harbin, Heilongjiang, China); Thiobarbital acid (TBA), guanidine hydrochloride, 2, 4-Dinitrophenylhydrazine (DNPH), trichloroacetic acid (TCA), Coomassie brilliant blue (CBB), bovine serum albumin (BSA) were from Shuanghuan Technology Development Co., Ltd. (Harbin, Heilongjiang, China). All other reagents used were analytical grade.
Q4: Line 85- Line 95: What was the level of salt and fat added to the burger?; How are hamburgers stored? in polystyrene trays? Did they wrap themselves in film?; use 12 h instead of 12 hours.
A4: Thanks for your review carefully. But our pork patties only were treated with the Pe powder for improving the stability of frozen pork patties storage. You are right. In the revised manuscript, the content you proposed have been corrected in line 87-100 as follow:
2.3. Preparation of pork patties
After thorough mixing the meat and incorporated ingredients (perform the same stirring operation on the blank group as the experimental group), shape the mixture in a mold (make a circular pork patties with a diameter of 6 cm and a thickness of 1 cm) and wrap themselves in film. Each group has 3 parallel samples, totaling 15 samples. The first freezing-thawing treatment of the test sample was to store it in a -20 ℃ refrigerator for 7 days, then the samples were thawed in a 4 ℃ refrigerator for 12 h. The 0, 1, 3, and 5 F-T treatments were performed in sequence. After each treatment ended, a certain amount of sample was used for the determination of relevant indicators.
Q5: Line 100- Line 105: Was any equipment used to homogenize the samples?; 5 g; 9000 rpm for 10 min; insert temperature of centrifugation process, as well as the equipment information (model, trademark, country)
A5: Thanks for your review carefully. You are right. In the revised manuscript, the content you proposed have been corrected as follow:
The sentences in line 100-105 have been corrected in line 100-111 as follow:
2.4. pH value
Pork samples (5 g) and 45 mL of sterile deionized water were mixed and homogenized by ultra high speed homogenizing emulsifier (Y25, Shanghai Yuedi Machinery Equipment Co., Ltd., Shanghai, China) for 60 s. Then, a precision pH meter (PHS-3E, Shanghai Yidian Scientific Instrument Co., Ltd., Shanghai, China) was used to determine the pH of the mixture [17].
2.5. Water holding capacity
2.5.1. Centrifugal loss
The pork samples were weighed to 5 g before centrifuging [18]. After the samples were centrifuged at 9000 rpm for 10 min at 4 ℃ by high speed centrifuge (HL-16KS, Shanghai Chuangbo Biotechnology Co., Ltd., Shanghai, China), the water on the surface of the samples was removed with degreased cotton, and the samples were weighed again.
Q6: Line 118- Line 136: 30 min; include equipment information for texture analyzer; 3 g; 30 s; 10 min; 3 s; 5 min; include equipment information for spectrophotomer
A6: Thanks for your review carefully. You are right. In the revised manuscript, the content you proposed have been corrected as follow:
The sentences in line 118-136 have been corrected in line 122-144 as follow:
2.6. Color and texture
After thawing, the sample was placed in air for reaction for 30 min, and measured using a colorimeter (ZE-6000, Shanghai Shouli Industrial Co., Ltd., Shanghai, China). The equipment was calibrated with a white board (L *=90.26, a *=-1.29, b *=5.18) to determine the L * value, a * value, and b * value of the pork patty. Using a texture analyzer (Brookfield CT-3, Shanghai Weichuan Precision Instrument Co., Ltd., Shanghai, China), the sample was steamed and cut into sections with a height of 20 mm and a diameter of 8 mm. A P/50 probe was used, with a pre test rate of 2.0 mm/s, a test rate of 2.0 mm/s, and a post test rate of 2.0 mm/s. The measurement mode was "strain" 40%, and 3 parallel samples were measured in each group, taking the average value.
2.7. Lipid oxidation and protein oxidation
2.7.1. Lipid oxidation
Peroxide value (POV) was determined with some modifications [14]. First, 3 g thawed minced meat was homogenized at high speed with 20 mL of chloroform methanol mixed solution for 30 s. Next, 3 mL of 0.5% NaCl solutio was added to the mixture and was centrifuged at 3000 rpm for 10 min. After 10 mL of the clear solution was taken and mixed with an equal volume of chloroform methanol mixed solution in glass test tube, 25 μL ammonium thiocyanate solution and 25 μL ferrous chloride solution were added successively, both vortexed for 3 s. Then the glass test tube was placed at room temperature for 5 min, and its absorbance was measured at 500 nm by Microplate reader (Epoch 2, Beijing Zhiyan Technology Co., Ltd., Beijing, China).
Q7: Line 139- Line 151: did you mean…17 mL of TCA solution were…; 30 min; 10 min; did you mean 3000 rpm; use the abbreviated form for bovine serum albumin, which was used at the beginning of this section. Note: it is recommended to only use the abbreviation after being mentioned in the text; 15 min; 5 min
A7: Thanks for your review carefully. You are right. In the revised manuscript, the content you proposed have been corrected as follow:
The sentences in line 139-151 have been corrected in line 147-159 as follow:
Next, 3 mL of TBA solution and 17 mL of TCA solution were added into the tube. The reaction solution was in a boiling water bath for 30 min after being mixed well. Finally, the obtained solution was centrifuged for 10 min (3000 rpm/min) at 4 ℃ and the absorbance was measured at 532 nm.
2.7.2. Protein oxidation
The degree of protein oxidation was analyzed by the method of protein sulfhydryl and carbonyl content determination [20]. Using BSA as the standard, the protein content was determined using the Coomassie brilliant blue method. 2.5 mL Tris Gly 8M urea solution was added to 0.5 mL of sample homogenization solution, shaking thoroughly to mix well duplicating two parts. 20 μL Ellman reagent was added to the experimental group; 20 μL Tris Gly solution was added to the control group. water bath at 37 ℃ for 15 min, centrifuge at 10,000 rpm at 4 ℃ for 5 min, and determine the absorbance value of the supernatant at 412 nm.
Q8: Line 151- Line 167: include temperature of centrifugation process; 3 g; 10,000 rpm; 1 min; 0.1 mL; 0.5 mL; 2 M; 0.02 M; 15 min; 0.5 mL; 10,000 rpm for 10 min; Also include the temperature of the centrifugation process; 1 mL; 6 M; 15 min; 26 h
A8: Thanks for your review carefully. You are right. In the revised manuscript, the content you proposed have been corrected as follow:
(1)The sentences in line 151-167 have been corrected in line 160-169 as follow:
In brief, 3 g of samples and 27 mL of physiological saline were placed in a 50-mL centrifuge tube. Then, the mixture was homogenized at 10,000 rpm for 1 min. Next, obtained 2 parts of 0.1 mL sample homogenization solution were respectively added into the following two groups: 1) 0.5 mL of 2 M HCl; 2) 0.5 mL of 2 M HCl (containing 0.02 M DNPH). After heated at 37 ℃ for 15 min in a water bath, the two groups were added 0.5 mL of 20% TCA solution and centrifuged at 10,000 rpm at 4 ℃ for 10 min to obtain the precipitate. The 2 parts of precipitates were washed three times with a mixture of alcohol and ethyl acetate, and centrifuge again at 10,000 rpm at 4 ℃ for 10 min. Finally, the 2 parts of precipitates were separated and both in 1 mL 6 M GuCl solution at 37 ℃ for 15 min,
(2)The sentences in line 151-167 have been corrected in line 174 as follow:
glutaraldehyde solution for approximately 26 h and dehydrated with a series of ethanol
Q9: Line 183- Line 261: did you mean Results and Discussion?; use [30-31] instead of [30-31]; …choice [reference?]. The addition of…
A9: Thanks for your review carefully. You are right. In the revised manuscript, the content you proposed have been corrected as follow:
(1)The sentences in line 183-261 have been corrected in line 190 as follow:
- Resultsand discussion
(2)The sentences in line 183-261 have been corrected in line 245 as follow:
increase water and fat retention in meat products [30-31], which greatly improve the
(3)The sentences in line 183-261 have been corrected in line 258-259 as follow:
Color, as an desirable attribute for meat products, often intuitively influences the consumer's choice [3].
Q10: Line 519- Line 650: insert space…Agric. Food; Chem.; Sci.; remove and; LWT.; use the abbreviated format for the journal name; Is the name of the journal correct? name change? use the correct abbreviation; LWT.; Crit.; study.; Note: use italic text format for scientific names in references section; Note: use the correct abbreviation in references section, J. Agric. Food Chem.
A10: Thanks for your review carefully. You are right. In the revised manuscript, the content you proposed have been corrected in lines as follow:
Line 521-522: Kong, C.H.Z.; Hamid, N.; Liu, T.T.; Sarojini, V. Effect of antifreeze peptide pretreatment on ice crystal size, drip loss, texture, and volatile compounds of frozen carrots. J. Agric. Food Chem. 2016, 64(21), 4327-4335.
Line 525-526: Wu, G.Y.; Yang, C.; Bruce, H.L.; Roy, B.C.; Li, X.; Zhang, C.H. Effects of alternating electric field during freezing and thawing on beef quality. J. Agric. Food Chem. 2023, 419.
line 527-528: Hu, R.; Zhang, M.; Jiang, Q.Y.; Law, C.L. A novel infrared and microwave alternate thawing method for frozen pork: Effect on thawing rate and products quality. Meat Sci. 2023, 198.
Line 531-532: Zahid, M.A.; Eom, J.U.; Parvin, R.; Seo, J.K.; Yang, H.S. Changes in quality traits and oxidation stability of syzygium aromaticum extract-added cooked ground beef during frozen storage. Antioxidants 2022, 11(3).
Line 537-538: Wang, X.; Xu, M.; Cheng, J.; Zhang, W.; Liu, X.; Zhou, P. Effect of Flamrnulina velutipes on the physicochemical and sensory characteristics of Cantonese sausages. Meat Sci. 2019, 154, 22-28.
Line 541-542: Sun, Y.N.; Hu, X.L.; Li, W.X. Antioxidant, antitumor and immunostimulatory activities of the polypeptide from Pleurotus eryngii mycelium. Int. J. Biol. Macromol. 2017, 97, 323-330.
Line 545-546: Wang, L.Y.; Li, C.; Ren, L.L.; Guo, H.Y.; Li, Y. Production of pork sausages using Pleaurotus eryngii with different treatments as replacements for pork back fat. J. Food Sci. 2019, 84(11), 3091-3098.
Line 547-548: Chung, S.I.; Kim, S.Y.; Nam, Y.J.; Kang, M.Y. Development of surimi gel from king oyster mushroom and cuttlefish meat paste. Food Sci. Biotechnol. 2010, 19(1), 51-56.
Line 551-553: Huang, L.; Liu, Q.; Xia, X.F.; Kong, B.H.; Xiong, Y.L.L. Oxidative changes and weakened gelling ability of salt-extracted protein are responsible for textural losses in dumpling meat fillings during frozen storage. J. Agric. Food Chem. 2015, 185, 459-469.
Line 556-557: Qing, Z.; Cheng, J.; Wang, X.; Tang, D.; Liu, X.; Zhu, M. The effects of four edible mushrooms (Volvariella volvacea, Hypsizygus marmoreus, Pleurotus ostreatus and Agaricus bisporus) on physicochemical properties of beef paste. LWT. 2021, 135.
Line 560-561: Shi, S.; Xu, X.W.; Feng, J.; Ren, Y.M.; Bai, X.; Xia, X.F. Preparation of NH3- and H2S-sensitive intelligent pH indicator film from sodium alginate/black soybean seed coat anthocyanins and its use in monitoring meat freshness. Food Packaging Shelf 2023, 35.
Line 565-566: Boonsumrej, S.; Chaiwanichsiri S.; Tantratian S.; Suzuki T.; Takai R. Effects of freezing and thawing on the quality changes of tiger shrimp (Penaeus monodon) frozen by air-blast and cryogenic freezing. J. Food Eng. 2007, 80(1), 292-299.
Line 575-576: Santos, E.M. Effect of the addition of edible mushroom flours (Agaricus bisporus and Pleurotus ostreatus) on physicochemical and sensory properties of cold-stored beef patties. J. Food Process. Pres. 2020, 44(3).
Line 584-585: Li, X.B.; Feng, T.; Zhou, F.; Zhou, S.; Liu, Y.F.; Li, W.; Ye, R.; Yang, Y. Effects of drying methods on the tasty compounds of Pleurotus eryngii. J. Agric. Food Chem. 2015, 166, 358-364.
Line 586-587: Jeong, J.Y.; Kim, G.D.; Yang, H.S.; Joo, S.T. Effect of freeze-thaw cycles on physicochemical properties and color stability of beef semimembranosus muscle. Food Res. Int. 2011, 44(10), 35.3222-3228.
Line 594-595: Gudbjornsdottir, B.; Jonsson, A.; Hafsteinsson, H.; Heinz, V. Effect of high-pressure processing on Listeria spp. and on the textural and microstructural properties of cold smoked salmon. LWT. 2010, 43(2), 366-374.
Line 598-599: Wang, X.P.; Zhou, P.F.; Cheng, J.R.; Chen, Z.Y.; Liu, X.M. Use of straw mushrooms (Volvariella volvacea) for the enhancement of physicochemical, nutritional and sensory profiles of Cantonese sausages. Meat Sci. 2018, 146, 18-25.
Line 602-604: Carneiro, C.d.S.; Marsico, E.T.; Resende Ribeiro, R.d.O.; Conte Junior, C.A.; Alvares T.S.; Oliveira de Jesus, E.F. Studies of the effect of sodium tripolyphosphate on frozen shrimp by physicochemical analytical methods and low field nuclear magnetic resonance (LF H-1 NMR). LWT. 2013, 50(2), 401-407.
Line 605-606: Chen, Q.; Huang, J.; Huang, F.; Huang, M.; Zhou G. Influence of oxidation on the susceptibility of purified desmin to degradation by mu-calpain, caspase-3 and-6. J. Agric. Food Chem. 2014, 150, 220-226.
Line 609-610: Kuribayashi, T.; Kaise, H.; Uno, C.; Hara, T.; Hayakawa, T.; Joh, T. Purification and characterization of lipoxygenase from Pleurotus ostreatus. J. Agr. Food Chem. 2002, 50(5), 1247-1253.
Line 620-621: Bach, F.; Ferreira Zielinski, A.A.; Helm, C.V.; Maciel, G.M.; Pedro, A.C.; Stafussa, A.P.; Avila, S.; Isidoro Haminiuk, C.W. Bio compounds of edible mushrooms: in vitro antioxidant and antimicrobial activities. LWT. 2019, 107, 214-220.
Line 628-629: Bao, H.N.D.; Ushio, H.; Ohshima, T. Antioxidative activity and antidiscoloration efficacy of ergothioneine in Mushroom (Flammulina velutipes) extract added to beef and fish meats. J. Agr. Food Chem. 2008, 56(21), 10032-10040.
Line 630-631: Abdullah, K.; Hüseyin, G. Enrichment of meat emulsion with mushroom (Agaricus bisporus) powder: Impact on rheological and structural characteristics. J. Food Eng. 2018, 237, 128-136.
Line 632-633: Zhang, W.; Xiao, S.; Ahn, D.U. Protein oxidation: basic principles and implications for meat quality. Crit. Rev. Food Sci. 2013, 53(11), 1191-1201.
Line 648-650: Lu, W.W.; Wu, D.; Wang, L.M.; Song, G.Y.; Chi, R.S.; Ma, J.; Li, Z.S.; Wang, L.; Sun, W.Q. Insoluble dietary fibers from Lentinus edodes stipes improve the gel properties of pork myofibrillar protein: A water distribution, microstructure and intermolecular interactions study. Foods 2023, 411.
The newly cited references:
- Pan N.; Dong C.; Du X.; Kong H.; Xia X.F. Effect of freeze-thaw cycles on the quality of quick-frozen pork patty with different fat content by consumer assessment and instrument-based detection. Meat Sci. 2020, 172(5), 108313.
- KimH.W.; Miller D.K.; Lee Y.J.; Kim, Y.H. Effects of soy hull pectin and insoluble fiber on physicochemical and oxidative characteristics of fresh and frozen/thawed beef patties. Meat Sci. 2016, 117, 63-67.

Round 2
Reviewer 1 Report
Comments and Suggestions for Authors
The authors responded to all my comments.
The paper has improved and can be published.